# International airport emissions and their impact on local air quality: Chemical speciation of ambient aerosols at Madrid-Barajas Airport during AVIATOR Campaign

Saleh Alzahrani[1], Doğuşhan Kılıç[1,2], Michael Flynn[1], Paul I. Williams[1,2] and James Allan[1,2]

[1]Department of Earth and Environmental Sciences, University of Manchester, Manchester, UK
[2]National Centre for Atmospheric Science, University of Manchester, Manchester, UK

*Correspondence to*: Saleh Alzahrani (Saleh.alzahrani@manchester.ac.uk)

**Abstract.** Madrid-Barajas International Airport (MAD), is the fourth-busiest airport in Europe. The aerosol chemical composition and the concentrations of other key pollutants were measured at the airport perimeter during October 2021, to assess the impact of airport emissions on local air quality. A high-fidelity ambient instrumentation system was deployed at Madrid Airport to measure: concentrations of organic aerosols (with their composition), black carbon ($e$BC), carbon dioxide ($CO_2$) carbon monoxide (CO), nitrogen dioxide ($NO_x$), sulphur dioxide ($SO_2$), particulate matter ($PM_{2.5}$, $PM_{10}$), total hydrocarbon (THC), and total particle number. The average concentration of $e$BC, $NO_x$, $SO_2$, $PM_{2.5}$, $PM_{10}$, CO and THC at the airport for the entire campaign were, 1.07 (µg/m³), 22.7 (µg/m³), 4.10 (µg/m³), 9.35 (µg/m³), 16.43 (µg/m³), 0.23 (mg/m³) and 2.30 (mg/m³) respectively. The source apportionment analysis of the non-refractory organic aerosol (OA) using positive matrix factorisation (PMF) allowed us to discriminate between different sources of pollution, namely: Less Oxidised Oxygenated Organic Aerosol (LO-OOA), Alkane Organic Aerosol (AlkOA), and More Oxidised Oxygenated Organic Aerosol (MO-OOA). The results showed that LO-OOA and MO-OOA accounts for more than 80% of the total organic particle mass measured near runway. Trace gases correlate better with AlkOA factor than LO-OOA and MO-OOA indicating that AlkOA is mainly related to the primary combustion emissions. Bivariate polar plots were used for the pollutant source identification. Significantly higher concentrations of the obtained factors were observed at low wind speeds (< 3m/s) from the southwest, where two of runways, and all terminals are located. Higher $SO_2/NO_x$ and CO/$e$BC ratios were observed when the winds originating from the northeast, where the two northern runways are located. These elevated ratios are attributed to the aircraft activity being the major source in the northeast area.

## 1. Introduction

Several studies have linked particulate matter (PM) to a range of harmful health effects, including respiratory and cardiovascular ailments (Boldo et al., 2006; Li et al., 2003a; Pope and Dockery, 2006; Schwarze et al.et al., 2006). In recent years, a number of researchers have found an association between aviation emissions and potential adverse human health impacts. These emissions can lead to immune system malfunction, various pathologies, the development of cancer, and premature death. Hence, it is increasingly recognised as a serious, worldwide public health concern (Yim et al., 2013; He et al., 2018; Jonsdottir et al., 2019). Airports contribute to primary and secondary inhalable and fine particulate matter ($PM_{10}$ and $PM_{2.5}$, with aerodynamic diameters of <10 µm and <2.5 µm, respectively), making them key determinants of urban air quality and a significant concern for local air quality management.

A few studies have reported that air pollutants emitted from large airports can play a vital role in worsening the regional air quality (Rissman et al., 2013; Hudda and Fruin, 2016). Hu et al., (2009) and Westerdahl et al., (2008) measured high ambient PM concentrations downwind of Los Angeles International Airport (LAX) and Santa Monica Airport (SMA) in California. A decline in the ambient air quality was observed up to 18 km downwind from international airports due to an increase in particle number concentrations linked to gas turbine-emitted PM (Hudda et al., 2014; Hudda and Fruin, 2016). To date, several questions still remain to be answered regarding the chemical composition of aircraft plumes, and the health risks associated with the exposure to the pollutants originating from airports in neighbouring communities. Responding to the growing concern about the risk of exposure to airport pollutants, studies have been conducted to gain a better understanding of airport emissions and their possible effects on local and regional air quality. Thus far, aircraft engines are considered to be one of the major sources of both gaseous and particulate pollutants at the airport (Masiol and Harrison, 2014). Various campaigns have reported both physical and chemical properties of particulate and gaseous emissions (Kinsey,

2009; Kinsey et al., 2010, 2011; Mazaheri et al., 2011; Hudda et al., 2016). Aviation fuel Jet A1 is the most common type of fuel that is used in civil aviation. It's a complex mixture of aliphatic hydrocarbons and aromatic compounds, characterized by a mean C/H ratio of $\sim$ 0.52 (with an average empirical molecular formula of $C_{12}H_{23}$) (Lee et al., 2010). The mass fraction of paraffins in jet fuel is over or equal to 75%, while the aromatic content is less than or equal to 25% (Liu et al., 2013). Although there are several fuel combustion sources at airports, including aircraft operation and diesel ground transport, the combustion of the aviation fuel increases maximum particle counts in the 10 - 20 nm range based on the particle size distribution analysis (Zhu et al., 2011). Other sources of airport-related PM emissions also contribute to local air pollution. Approximately 38% of $PM_{10}$, with a mean level of 48 µg/m³ at airports, can periodically originate from the construction activities related to terminal maintenance and expansion (Amato et al., 2010). Particles emitted by commercial aircraft can be divided into two main groups: non-volatile and volatile PM. Non-volatile PM (nvPM) is usually formed during the (incomplete) combustion process and then emitted from the aircraft combustion chamber. It consists mostly of carbonaceous substances such as soot, dust, and trace metals (Yu et al., 2019). nvPM has the physical property of being resistant to high temperatures and pressure. On the other hand, volatile PM is formed through gas to particle conversion process, primarily by sulphur and organic compounds, which exist in the exhaust gas downstream of the engine after emission. Sulphuric compounds are formed as a result of sulphur in fuel, whereas organic particles are formed as combustion products, and from fuel and oil vapours (ICAO, 2016; Smith et al., 2022). Aircraft and ground unit emissions have been documented in prior research (Masiol and Harrison, 2014), yet there is still a gap in knowledge about airport-related PM emissions in terms of (i) apportioning PM to individual sources at airports, (ii) specifying their chemical composition, and (iii) the wider impacts of PM on local communities. This study aimed to obtain data to address these research gaps by providing further in-depth information on particle composition measurements and key pollutants observed within an airport environment. It characterises organic volatile PM emissions to assess the effect of aviation emissions on the local air quality. As part of the AVIATOR Project (Assessing aViation emission Impact on local Air quality at airports: TOwards Regulation), ambient measurements were conducted at Madrid–Barajas Airport to monitor the chemical properties of sub-micron particles near the runways. Source apportionment analysis was performed based on the particle data collected via high resolution mass spectrometry and this analysis allowed us to discriminate between different sources of air pollution at the airport microenvironment. These findings will serve as the foundation for additional comprehensive research, such as toxicological and health effect studies of PM originating from aviation activities.

## 2. Methods
### 2.1. Description of the sampling location

Adolfo Suárez Madrid-Barajas Airport is the main international airport in Spain, located within the municipal limits of Madrid, 13 km northeast of Madrid's city centre. It is the fourth-busiest airport in Europe based on passenger volume (Eurostat Database, 2021). In 2019, 62 million travellers used Madrid-Barajas and nearly half a million aircraft movements have been recorded, making it the largest and busiest airport in the country. In 2021, nearly one-third of the previous number travelled through Madrid Airport because of the COVID-19 pandemic. The airport has five passenger terminals named T1, T2, T3, T4, and T4S. Barajas Airport also has four runways: two on the north-south axis, parallel to each other (18L/36R and 18R/36L), and two on the northwest-southeast axis (14L/32R and 14R/32L). The runways enable simultaneous takeoffs and landings at the airport, allowing 120 operations per hour (one takeoff or landing every 30 seconds). The sampling location was chosen in collaboration with AENA, the owner and operator of the Barajas Airport, to facilitate the provision of power and access for servicing. Focusing on the temporal and spatial monitoring of the key pollutants, the site was positioned between runways 36L and 36R to sample the airport emissions from an optimal sampling point for aviation activities (Fig.1). The distance from sampling location to the runways 18L/36R, 18R/36L, 14L/32R and 14R/32L are 680 m, 620 m, 3.2 km, and 4.1 km respectively. Furthermore, the distance between sampling location and adjacent terminals T1, T2, T3 is approximately 5 km whereas 3 km and 1.5 km to the terminals T4 and TS4 respectively. The nearest highway is located around 2.6 km away from the sampling location.

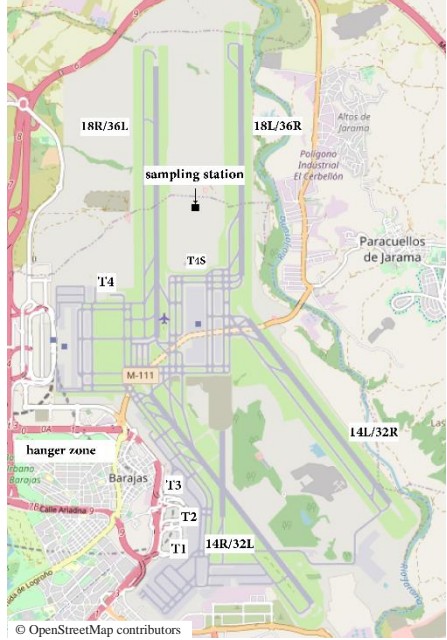

© OpenStreetMap contributors

**Figure 1. Locations of runways, terminals, and sampling site at Adolfo Suárez Madrid-Barajas Airport. Measurements were performed between October 8, 2021 and October 23, 2021. (Adapted from: https://www. openstreetmap.org)**

## 2.2. Sampling and instrumentation

The autumn campaign of AVIATOR took place in October 2021. Sampling was conducted continuously, starting at 12:00 pm on October 8, 2021 and ending at 20:00 pm on October 23, 2021. An ambient instrumentation system with specific reference to PM was deployed at Madrid Airport to better characterise air quality at the airport microenvironment. The measurement equipment of the system includes an Aerodyne High-Resolution Time-of-Flight Aerosol Mass Spectrometer (AMS) for the chemical speciation of the particles. AMS measures concentration and chemical composition of non-refractory aerosols online. AMS provided high-resolution measurements of primary and secondary organic aerosol and inorganic aerosol including sulphates, nitrates, and ammonium, from approximately 60 nm to 600 nm with 100 % transmission, extending to smaller and larger sizes with reduced transmission (Canagaratna et al., 2007). An aerodynamic lens is used to draw aerosols into a vacuum chamber. Particles are focused into a narrow beam and accelerated to a velocity inversely related to their vacuum aerodynamic diameter. The particles impact on a tungsten surface, heated to 600 °C, which causes them to flash vaporise. A 70-eV electron is used to ionize the vapours before they are analysed by mass spectrometry. During the measurement period, AMS was sampling with 1μm cut-off inlet and at 30 s time resolution. In addition to standard AMS flow, baseline and single ion calibrations every second day, an ammonium nitrate solution was atomised to calibrate the AMS (for size-dependent ionisation efficiency). The analysis of the chemical characteristics of aircraft PM using an AMS have been described elsewhere in detail (Yu et al., 2010; Anderson et al., 2011; Smith et al., 2022). Equivalent black carbon mass concentration ($e$BC) based on aerosol optical absorption was monitored using the Multi-Angle Absorption Photometer (MAAP) during this campaign. The MAAP operates at 670nm wavelength, has a 10s-time response with a flow rate of 8 litre/min, for unattended long-term monitoring of carbonaceous particulate emissions from combustion sources (Petzold and Schonlinner, 2004). MAAP has been used for the monitoring of black carbon emission from aviation (Herndon et al., 2008; Timko et al., 2014). The instrument was set up to measure average $e$BC concentrations with one-minute intervals. By using a condensation particle counter (CPC), TSI model 3750 ($D_{50}\approx$7nm), total particle number concentration was measured real-time to capture temporal variability in particle number concentrations with a measurement range of up to 100,000 particles/cm³ and a time resolution of one second. Ambient $CO_2$ concentration near runways were also measured by a LI-COR $CO_2$ Trace Gas Analysers at 1-sec intervals. In addition, meteorological parameters (temperature, pressure, relative humidity, wind speed, and direction) were measured at the site with the instrumentation system. The system was co-located with AENA (REDAIR) fixed monitoring site to provide additional spatially resolved data. The REDAIR station monitors the concentration of sulphur dioxide ($SO_2$), nitrogen dioxide ($NO_x$), carbon monoxide (CO), ozone ($O_3$), suspended particles PM (including $PM_{2.5}$, $PM_{10}$), and total hydrocarbon (THC) with a time resolution of 30 minutes.


**2.3. Data analysis**

AMS was operating in Mass Spectrum (MS) mode to identify the chemical species present in the aerosol ensemble
and quantify the overall mass loading. AMS data were analysed using the data analysis toolkit TOF-AMS
SQUIRREL v1.65B, operated within Igor Pro (WaveMetrics, Inc.). The Source Finder (SoFi) is a software
package designed to analyse multivariate data using state-of-the-art source apportionment techniques to
understand the sources of various pollutants (Canonaco et al., 2013). SoFi, running under IGOR 6.37, was used
to deconvolve organic aerosol emissions via the Positive Matrix Factorization (PMF) model. The PMF model,
implemented through the multilinear engine version 2 (ME-2) factorisation tool, was used to determine the number
of factors (sources). ME-2, a multivariate solver, employs the same mathematical/statistical method as PMF to
evaluate solutions (Paatero, 1999). ME-2 equations are designed for analysing and calculating the relative
contributions of various source pollutants by measuring their concentration at receptor locations (Paatero and
Tapper, 1994). The PMF model processes many variables and categorises them into two types (i) source types,
which can be determined based on the chemical composition of the pollutants, and (ii) source contributions, used
to quantify the amount of contribution from each source to a sample. PMF inputs were restricted to only non-
negative concentrations since no sample can have a negative source contribution. A step-by-step approach was
employed to select the number of solutions (factors). The method described by Reyes et al. (2016) and Smith et
al. (2022) was used to determine the optimal solution. This approach began initially with a two-factor model and
then incrementally increased to a maximum of five factors. PMF analysis was performed with seed runs and
varying FPEAK values (ranging from -1 to 1 with steps of 0.1) to better differentiate organic aerosol sources.
Seed runs and FPEAK are rotational techniques in the ME-2 tool, and they represent one of the unconstrained
PMF run approaches used for the exploration of the solution space. During the analysis, it was noted that factor
four consistently correlates with factor five, exhibiting identical time series and similarities in mass spectra. This
difficulty in separation has previously been observed in the case of well-mixed pollutants, attributed to low
temperatures and wind speeds (Reyes et al., 2018). Greater stability was achieved when analysing 3-factor
solutions with varying FPEAK values. During the analysis, seed runs and PMF with FPEAK solutions showed no
significant variation in the normalised scaled residuals parameter (Q / Qexp), with values close to 1. This is
reasonable given that PMF determines the solution by minimising this value (Reyes et al., 2016). The factorisation
strategy was entirely successful in separating three different sources, each with distinct mass spectra and differing
time series. Consequently, 3-factor solutions emerged as the optimal number of sources, demonstrating the best
performance with the lowest residuals and Q/Qexp values close to 1. Furthermore, the obtained solution exhibited
the most favorable results, characterized by distinct diurnal trends and dissimilarities in time series and mass-to-
charge ratios among the factors.


**3. Results and Discussion**
**3.1 Variations of organic, inorganic, and oil emissions**

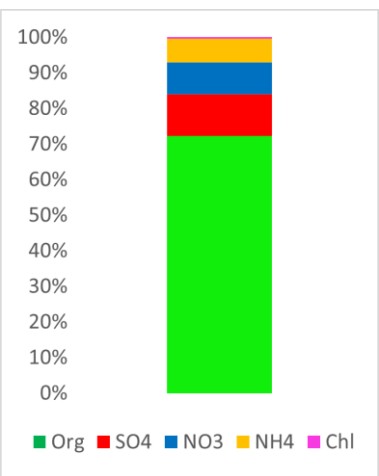


**Figure 2. The bar chart shows aerosol fractions where organic and sulphate species account for more than 80% of the**
**total aerosol mass.**

The average mass concentration of organic and inorganic aerosols during the entire campaign was 9.6 µg/m³. The
bar chart in Fig. 2 shows aerosol fractions, with organic species accounting for more than 70% of the total aerosols.
This is significantly higher than the nearest component, sulphate, which accounted for 15%. It should also be

noted that the nitrate and sulphate species measured by AMS could potentially contain an organic fraction. The PMF analysis in this paper primarily focuses on the composition of the organic mass concentration, which is discussed in further detail in Section 3.2. Previous studies have shown that lubrication oil has been detected in ambient air near runways, and it may further add to the total organic PM emissions due to aircraft engine operations (Timko et al., 2010b; Yu et al., 2010; Fushimi et al., 2019; Ungeheuer et al., 2022). Aircraft plume measurements indicated that oil was found to contribute 5% to 100% (Yu et al., 2012). The m/z 85 signal is a well-known oil marker in the AMS mass spectrum, attributed to synthetic esters ($C_5H_9O^+$) (Timko et al., 2014). Ratio of *m/z* 85:71 is used as a marker for oil (Fig. 3). The value of 0.66 was used as a benchmark for oil contribution (Yu et al., 2012). Values below 0.66 indicate oil-free organic PM, while values above 0.66 suggest the presence of lubrication oil. However, based on the AMS measurements during AVIATOR autumn campaign, lubrication oil accounted only up to 5% of the total aerosol mass, which is significantly less compared to the measurements of Yu et al. (2012). There are three probable explanations on the deficiency of AMS to detect oil precursors: (i) the oil particles are too small in diameter for AMS to detect, (ii) complete pyrolysis of the oil in the engine combustion zone forming carbon monoxide (CO) and carbon dioxide ($CO_2$) ( Smith et al., 2022) or (iii) oil particles contribute to an insignificant amount (by mass) to the organic mass in engine exhaust and therefore are not detected. Additional factors that could potentially impact the minimal presence of oil lubrication in this analysis might involve the overall mass loading of aerosols, the influence of urban aerosol emissions, or the proximity of the sampling point to the nearest runways. Additional information on how the lubrication oil, as measured by AMS, varies with wind speed and direction is provided in the supplementary material (Fig.S4). During the AVIATOR autumn campaign, measuring oil was challenging due to the prevalent urban background. A "little oil" region was identified at low to moderate wind speeds (2~5 m/s) originating from the southwest, encompassing terminal buildings (T1, T2, T3, T4, and TS4), two runways (14R/32L and 18R/36L), and a hangar zone. In contrast, a region "unlikely to contain oil" was noted when winds came from the northeast of the airport, near runways 18L/36R, with relatively higher wind speeds (above 5 m/s). Furthermore, Fig.S5 displays the daily ratio of *m/z* 85:71 throughout the sampling period, pinpointing Sunday, October 16th, as the only day when the oil marker surpassed 0.66. On other days, the ratio of *m/z* 85:71 suggested a minimal likelihood of oil presence. An hourly analysis within Fig.S5 reveals that the oil marker exceeded 0.66 only at 20:00, aligning with the evening peak in PM$_{2.5}$ concentrations Fig.S3. This suggests a significant influence of urban background aerosols on the lubrication oil measurements. Since PMF analysis is based on the organic masses measured via AMS, lubrication oil is not identified as a determinant and there is no oil organic mass profile reported in previous studies and here (Ulbrich et al., 2009). PMF has been proven inefficient at detecting such levels (Ulbrich et al., 2009), therefore, oil contribution to the organic mass may be under-represented in this study.

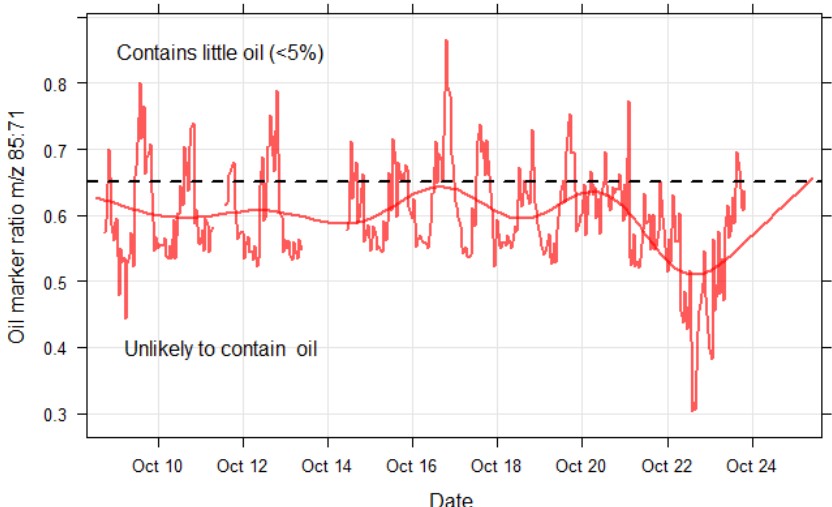

**Figure 3. Temporal variability of lubrication oil fraction in total aerosol mass obtained from AMS measurements. The ratio of m/z 85/71 was used as the mass marker to identify lubrication oil. A smooth red line is fitted to the data, while the dashed black line represents the value of 0.66, assumed for oil-free organic PM emitted from aircraft engines. The analysis showed that no oil or very little (<5%) oil fraction was detected during the measurement period.**

**3.2 PMF Analysis**

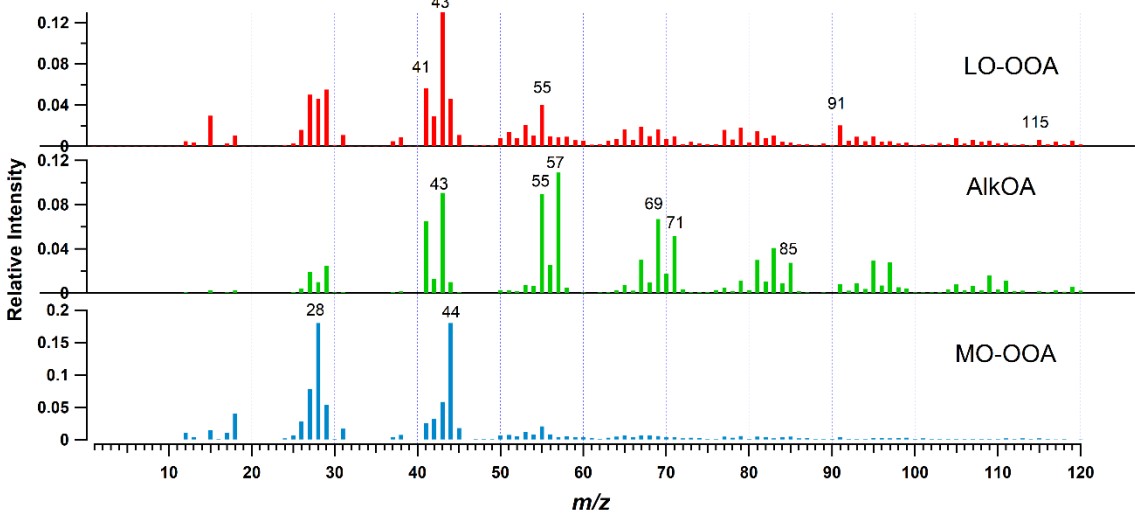

**Figure 4. The mass spectral fingerprint of the three factors from the PMF solutions. Less Oxidised Oxygenated Organic Aerosol (LO-OOA), Alkane Organic Aerosol (AlkOA), and More Oxidised Oxygenated Organic Aerosol (MO-OOA), which can be indicative of secondary aerosols. Selected mass markers with a relative intensity higher than 0.01 are numbered.**

The PMF analysis in this study aims to provide relative contribution of the sources of aerosols near runway. In addition to determining the diurnal pattern of the obtained factors during the autumn campaign, PMF solutions were used to investigate how meteorology affects airborne particulate pollution. During AVIATOR autumn campaign at Madrid-Barajas International Airport three sources were identified via PMF (Fig. 4 shows the results of the PMF analysis, the mass spectral fingerprint). The first factor in Fig. 4, LO-OOA, stands for Less Oxidised Oxygenated Organic Aerosol. It is a type of secondary organic aerosol (SOA) characterized by its low degree of oxidation. LO-OOA are formed in the atmosphere through the oxidation of volatile organic compounds (VOCs), which can originate from a variety of anthropogenic sources. In this analysis LO-OOA shows the presence of an aromatic marker at $m/z$ 115, a marker used for identifying indene ($C_9H_8$) ion in previous studies focusing on aviation emissions (Timko et al., 2014; Smith et al., 2022).LO-OOA is associated with aromatic fragments at m/z 77 ($C_6H_5^+$), and 105 ($C_8H_9^+$). It presents a high relative intensity (0.13) at m/z 43 ($C_3H_7^+$) (characteristic of LO-OOA) and a lower relative intensity (<0.04) at m/z 91, which is related to toluene ion ($C_7H_7^+$) (Timko et al., 2014; Smith et al., 2022). Ambient temperature plays a crucial role in influencing the LO-OOA factor, displaying significant diurnal fluctuations. The lowest concentrations of LO-OOA are recorded at midday, coinciding with the peak in ambient temperatures (Fig. 5). A prior PMF analysis of organic particulate matter from aircraft emissions revealed a significant aromatic factor within the organic PM, characterized by elevated signals at $m/z$ 77, 91, 105, 115, 128 (Timko et al., 2014). The aromatic factor identified by Timko et al. (2014) was found to dominate the organic PM emissions from turbojet engines at low-thrust settings. It was associated with the products of incomplete combustion and exhibited high variability, which varied with engine power settings (the sum of signals in the factor decreased as engine power increased). Another study by Smith et al. (2022), investigated the chemical composition of organic aerosols emitted by gas turbines and identified a Semi-Volatile Oxygenated Organic Aerosol (SV-OOA) factor, which forms through oxidative processes near the engine exit. A strong correlation (R = 0.91) and similarity in mass spectra between the LO-OOA in this study and the SV-OOA described by Smith et al. (2022) were observed. Owing to the absence of volatility measurements during this period and the limited time for aging (no more than a few minutes), we consider the LO-OOA factor in our analysis to be the most accurate estimate available, rather than the SV-OOA as suggested by Smith et al. (2022). The second factor, identified based on the PMF analysis of Madrid airport sample, is Alkane Organic Aerosol (AlkOA) factor. It is associated with unburned fuel and emissions from incomplete combustion, exhibiting high relative intensities at $m/z$ 43, 57, and 85, indicative of decane ($C_{10}H_{22}$), a common alkane in jet fuel. Given that mass spectral fingerprint of decane is similar to the other aliphatic hydrocarbons (*e.g.*, long-chain alkanes) found in Jet A1 fuel, as reported by Yu et al. (2012) and Smith et al. (2022). AlkOA factor referred here as a marker to identify emissions originating from unburnt fuel/incomplete fuel combustion products. Previously, primary aliphatic factor was found in PMF analysis by Timko et al. (2014) and was characterized by increased signals at masses such as 41/43, 55/57, 69/71, 83/85. Each of these masses correspond to an alkane. The primary aliphatic factor in Timko et al. (2014) study was strongly correlated with black carbon soot emissions under high-power conditions. The

strong association between the primary aliphatic factor and soot emissions suggests they originate from similar
combustion processes. Timko et al. (2014) concluded that the primary aliphatic factor is derived from combustion
related sources and can potentially contain significant amounts of unburnt jet fuel. Additionally, a strong positive
linear correlation was observed between the AlkOA factor identified in this study and the decane factor from
NIST webbook (R= 0.83) (NIST Mass Spectrometry Data Center, 1990), as well as between the AlkOA factor
determined here and the AlkOA factor reported by Smith et al. (2022) (R=0.93). The positive linear correlation
among these three factors suggests they are indicative of similar primary pollutants derived from fuel vapours or
incomplete combustion products associated with jet fuel. Results are consistent with previous findings of another
study (Smith et al., 2022). The third factor, More Oxidised Oxygenated Organic Aerosol (MO-OOA), is a type of
secondary organic aerosol (SOA) that can form from various origins and processes, such as photochemical
processing of aged SOA and the regional-scale transport of chemical reactions. MO-OOA has a spectral
fingerprint that consists of more oxidised ions (compared to LO-OOA and AlkOA), indicating a secondary aerosol
fraction in the sample. MO-OOA is characterized by its notably high relative intensities (>0.18) at m/z 29 ($CHO^+$)
and 44 ($CO_2^+$), which serve as markers for its identification (Alfarra et al., 2007). Given that MO-OOA has the
highest *f44/43* ratio among the three factors, it is expected to be the most oxygenated (in terms of chemical content)
factor. Being more oxidised potentially makes MO-OOA less volatile than LO-OOA (Jimenez et al., 2009;
Smith et al., 2022). MO-OOA in this analysis indicates the formation of aged secondary organic aerosols with no
significant diurnal variation (Fig. 5), often associated with air masses transported from polluted regions. Other
sources may have been included in one or both factor solutions, consequently, this does not rule out the possibility
of their existence.
**3.3 The temporal distribution of factors and correlation with trace gases**

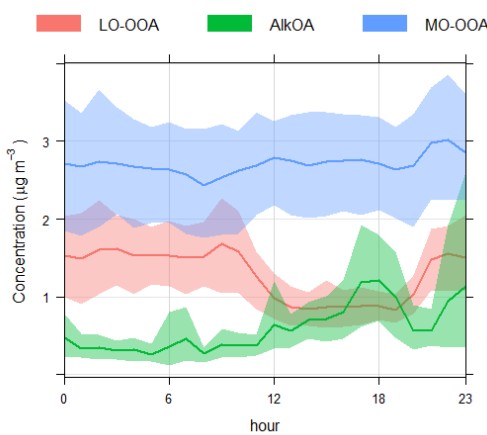

**Figure 5. Diurnal pattern of the solved factors from October 8, 2021 to October 23, 2021. The mean diurnal pattern is**
**shown as solid lines, and the shading indicates the 95% confidence interval around the mean.**
Average hourly concentrations of the PMF-determined factors were calculated based on the hourly organic aerosol
concentrations throughout the entire campaign to monitor the diurnal variation of the source contributions. The
variation of the AlkOA concentration during the day mostly associated with aircraft emissions (Fig. 5). The
concentration of AlkOA factor is relatively higher in the afternoon compared to the morning and midday. The
pattern of diurnal AlkOA closely resembles that of diurnal flight activities, suggesting that the surge in AlkOA
levels beginning at noon is linked to primary particles released by aircraft. The AlkOA factor shows an increase
between 09:00 and 20:00 and again between 22:00 and 23:00. Based on the mean diurnal pattern with a 95%
confidence interval, the AlkOA factor increases during the 09:00 to 20:00 period, corresponding with peak flight
activity (approximately 71% of total flights). Further details on daily aircraft activities are provided in the
supplementary material (Fig. S2). The increase in AlkOA between 22:00 and 23:00 is not statistically significant
due to high variability (Fig. 5). The increase in AlkOA concentration from 22:00 to 23:00, or the subsequent
decrease from 23:00 to 00:00, falls within the variability range of the 00:00 to 01:00 period. Therefore, a
statistically significant decrease in AlkOA concentration from 23:00 to 00:00 is hardly measurable.
Meteorological factors may contribute to the variability in the diurnal cycle observed during this period.
Additionally, unidentified local source such as airport ground service equipment could potentially explain the
variability observed from 22:00 to 00:00. This source has been previously reported as the main determinant of the
air quality in the vicinity of the airport (Masiol and Harrison, 2014). The LO-OOA factor likely represents fresh
secondary organic aerosols (SOA), demonstrating high variability and sensitivity to ambient temperature
fluctuations. The concentration of LO-OOA is at its lowest when daytime temperatures peak. LO-OOA may
contain urban contributions and potentially effected by background urban pollution from Madrid. The observed
reduction in LO-OOA factor during the afternoon can be attributed to dilution effects resulting from the rise in
boundary layer height, along with the potential evaporation of LO-OOA particles due to increased ambient
temperatures. This is supported by the variance in background particulate matter concentrations located south of
the airport compared to those at the sampling point, approximately 6 km apart, as illustrated in Fig. S3. (Fig. S3)
reveals that $PM_{2.5}$ levels at both locations experience significant increases during morning and evening rush hours,
with the sampling point consistently showing higher concentrations than the background location. The diurnal
pattern of the background location demonstrates a rapid decrease in $PM_{2.5}$ levels in the afternoon, unlike the
measurements at the sampling point. Additionally, there is a noticeable lag of about an hour between the peak
concentrations at the sampling point and those in the background, suggesting the influence of additional
combustion sources of $PM_{2.5}$, notably aviation-related activities, particularly during periods of increased airport
traffic. Unlike other factors, MO-OOA shows no significant diurnal variation, indication the formation of aged
secondary organic aerosols, often a result of atmospheric transport (Zhang et al., 2007). Detailed statistics of the
obtained factors for the entire campaign are provided in the supplementary material (Table S1). At Madrid-Barajas
Airport, AlkOA exhibited moderate correlations with $e$BC, $NO_x$, $SO_2$, and CO, as evidenced by the linear
correlation coefficients listed in Table 1 (R=0.56, R =0.52, R =0.53, and R =0.52). In contrast, the correlation of
these trace gases and both LO-OOA and MO-OOA is lower compared to AlkOA, with R values ranging from 0.2
to 0.5, as shown in (Table 1). The slightly higher correlation of AlkOA with BC, $NO_x$, $SO_2$ and CO (R > 0.5)
relative to LO-OOA and MO-OOA can be attributed to AlkOA being a primary pollutant, emitted directly from
the source. Conversely, LO-OOA and MO-OOA are believed to be secondary pollutants, formed through the
processes of condensation and coagulation of primary pollutants. In this study, urban contributions are
predominantly subject to this processing, as there is insufficient time for significant photochemical oxidation of
aviation emissions in such close proximity to the source. Additionally, the diurnal trends of BC, $NO_x$, $SO_2$ and
CO can be significantly affected by meteorological conditions (*e.g.*, wind speed, temperature) (Carslaw et al.,
2006; Reyes et al., 2018). This influence accounts for their moderate correlation with AlkOA, with R values
between 0.52 and 0.56, as detailed in Table 1. Similarly, AlkOA could potentially be affected by meteorological
conditions. Since AlkOA is measured as part of AMS sub-micron particles, it is expected to behave similarly to
$e$BC in the particle phase. Therefore, meteorological conditions likely influence both AlkOA and $e$BC in a similar
manner. AlkOA and trace gases were normalised to facilitate comparison of their diurnal patterns, thereby
enhancing understanding of their relative contributions and identifying trends among these pollutants.
Normalising is accomplished by dividing the concentrations of the pollutants by their average value. Figure 6
shows diurnal patterns of AlkOA factor, $e$BC, $NO_x$, CO, and particle number concentration. The daily trend of
$e$BC, $NO_x$ and CO are mostly similar, with very pronounced increases in concentrations during the morning and
evening rush hours. The average concentrations were 1.07 μg/m³, 22.7 μg/m³ and 0.23 mg/m³ for $e$BC, $NO_x$ and
CO respectively (Table S1). AlkOA gradually increases during the morning, with multiple minor peaks observed
in the morning hours. The average concentration of AlkOA is higher at night than during the day. This increase
is potentially related to daily aircraft activities. AlkOA began to increase, reaching a maximum during the
afternoon rush hour from 12:00-18:00. a second rapid increase occurred around 20:00, potentially caused by an
increase in the number of flights at this time (Fig. S2). Early morning AlkOA concentrations are significantly
lower compared to those of $e$BC, $NO_x$ and CO. This difference could be attributed to reduced emissions resulting
from decreased aircraft activities in early mornings (Fig. S2). The rise in trace gases and $e$BC observed in the
early morning hours could originate from various airport operations. Such operations might encompass emissions
from auxiliary power units, vehicle traffic, and the use of ground service equipment at the airport (Masiol and
Harrison, 2014). The total number concentration exhibited a temporal pattern similar to that of AlkOA from
15:00–21:00. Likewise, the temporal profiles of AlkOA and trace gases were similar during the afternoon period
(17:00-21:00). This similarity in temporal profiles suggests common source origins, which may be temporally
associated with aircraft activity or the influence of background urban pollution.
**Table 1 Results of linear regression analysis between obtained factors (LO-OOA, AlkOA, and MO-OOA)**
**and external tracers. Data from the entire campaign was used to perform the correlation analysis.**

|  | $e$BC (μg/m³) | $NO_x$ (μg/m³) | $SO_2$ (μg/m³) | CO (mg/m³) | THC (mg/m³) | $PM_{2.5}$ (μg/m³) | Tot No. conc (particles/cm³) | $CO_2$ (ppm) |
|---|---|---|---|---|---|---|---|---|
| LO-OOA | 0.49 | 0.28 | 0.21 | 0.32 | 0.63 | 0.36 | -0.08 | 0.24 |
| AlkOA | 0.56 | 0.52 | 0.53 | 0.52 | 0.35 | 0.66 | 0.4 | 0.35 |
| MO-OOA | 0.48 | 0.36 | 0.26 | 0.45 | 0.41 | 0.55 | 0.1 | 0.22 |


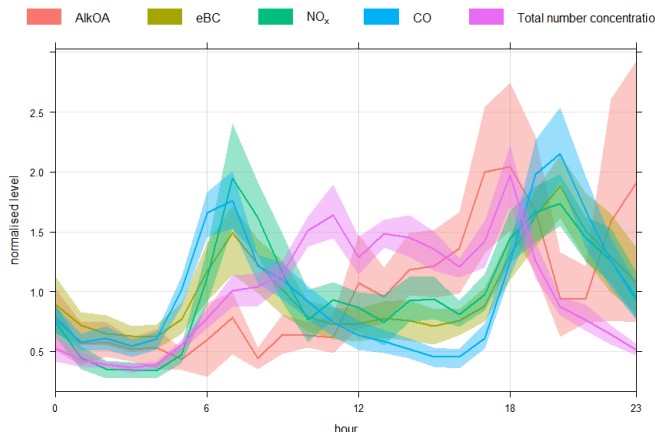

**Figure 6. The diurnal cycle of AlkOA compared to *e*BC, NOₓ, CO, and total number concentration. In this plot, the concentrations are normalised with the objective of comparing the patterns of different pollutants using the same scale.**

**3.4 Spatial analysis**

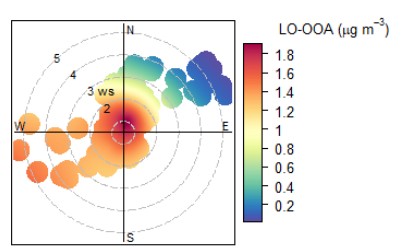 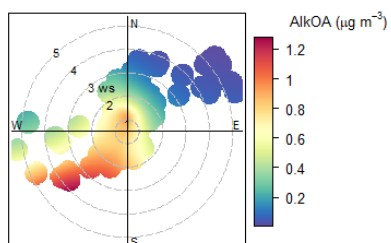 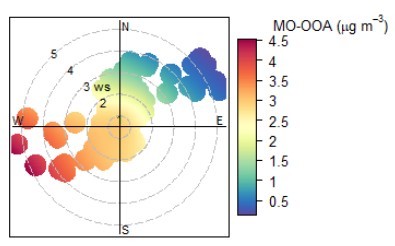

**Figure 7. Bivariate polar plots for LO-OOA, AlkOA, and MO-OOA (µg/m³). The highest concentrations were measured when the winds were originated from the west and southwest. Runways 18R/36L and 14R/32L located at western and eastern sides of the measurement station and the hanger zone with terminals T1, T2, T3, T4, and TS4 are located at the south and southwest of the measurement site (Fig. 1).**

Varying sources can be discriminated by means of bivariate polar plots techniques (Carslaw and Ropkins, 2012). Figure 7 illustrates the impact of airport activities on the average concentrations of factors (LO-OOA, AlkOA and MO-OOA) as determined by PMF. The highest concentrations of AlkOA and MO-OOA were observed at low to moderate wind speeds (3~5 m/s) coming from the west and southwest (R= -0.35 and R= -0.42, respectively), near the terminal buildings (T1, T2, T3, T4 and TS4), two of the runways (14R/32L and 18R/36L), and a nearby hanger zone. The most significant contributions of LO-OOA occur at wind speeds below 2 m/s, with a correlation of R= -0.45. At such low wind speeds (< 2 m/s), LO-OOA and MO-OOA are more likely to be mixed and influenced by a nearby source (Crilley et al., 2015; Helin et al., 2018). By contrast, the minimum significant contribution from all factors was observed when the winds originated from the northeast of the airport, accompanied by relatively higher wind speeds (above 4 m/s). Thus, based on the polar plots shown in Fig. 7, emissions from the terminal buildings and hanger zone located at the southwest of the measurement station are the major sources of total organic particle concentrations at the measurement station. The average contributions of LO-OOA, AlkOA, and MO-OOA were 1.63, 0.63, and 2.35 µg/m³, respectively (Table S1). During the AVIATOR campaign in October 2021, LO-OOA and MO-OOA constituted more than 80% of the total organic mass. Based on the strength of the relationship outlined in Table 1 between derived factors and external tracers, the linear correlations (Pearson correlation) between (i) AlkOA with *e*BC and (ii) LO-OOA with THC were measured under varying wind speed and directions, as illustrated in (Fig. 8). The relative contributions of the AlkOA and LO-OOA were higher with winds originating from southwest of the airport, compared to when winds carried air parcels to the sampling point from the northeast, as discussed. However, the correlation coefficient for these factors varies significantly, ranging from 0.2 to 0.9, for all samples collected from various directions within the airport perimeter. For instance, AlkOA

exhibits a strong linear correlation with $e$BC (Pearson coefficient higher than 0.9) when winds originate from the
west, east, or northeast, as illustrated in Fig. 8. This correlation is attributed to the impact of runways 18L/36R
and 18R/36L, which are situated to the east and west of the measurement site, respectively, as depicted in Fig. 1,
where 90% of aircraft take-offs occur. Both AlkOA and $e$BC are related to jet fuel emissions, as they are directly
emitted by aircraft engines as a result of fuel combustion. $e$BC emissions are a function of engine power settings,
reaching their maximum at full thrust during take-off (Kinsey et al. 2011; Hu et al., 2009). Furthermore, a
significant linear correlation was measured between LO-OOA and THC when dominant winds were north
easterlies (the air parcels move from runways 18L/36R to the sampling station). THC emissions at airports
primarily dependent on the jet engine thrust setting (Anderson et al., 2006; Onasch et al., 2009). When engines
operate at low thrust settings (*e.g.*, during landing, taxiing, idling), combustion is less efficient, leading to the
emission of higher amounts of hydrocarbons. The association between LO-OOA and THC in certain areas of the
airport can be interpreted as indicative of fresh emissions from aircraft in service.

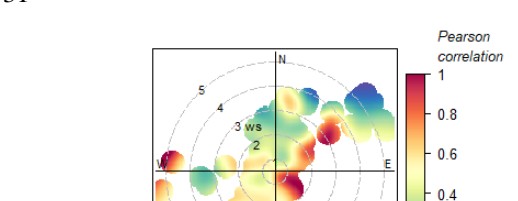 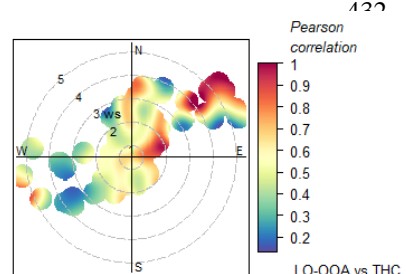

**Figure 8. A Pearson correlation analysis using bivariate polar plots (above) shows a significant positive linear**
**correlation between AlkOA with $e$BC and LO-OOA with THC mass concentrations when prevailing winds were**
**northeast. (The location of runways 18L/36R).**

$NO_x$ emitted by aircraft can potentially affect air quality up to 2.6 km away from the airport (Carslaw et al., 2006).
However, accurately determining the airport's contribution to local $NO_x$ concentrations presents challenges due
to other predominantly mobile sources of $NO_x$ in urban areas. In this study, the potential contribution of road
traffic surrounding the airport, particularly from the motorways located to the south and southwest, originates
from the same direction as runway 14R/32L and all the terminals. Therefore, $NO_x$ contributions were higher from
the south and southwest of the airport (including local on-road $NO_x$) compared to the those from the northeast.
The lowest $NO_x$ concentrations were measured under moderate wind speed conditions (above 4 m/s), as shown in
Fig. S1. This is possibly due to atmospheric mixing and plume dilution caused by advection (Carslaw et al., 2006),
given that ground-level source emissions are inversely proportional to wind speed. During this campaign, the
AENA (REDAIR) station located at the airport provided measurements of sulphur dioxide ($SO_2$) and carbon
monoxide (CO) (Fig. S1). Aviation activities have previously been reported as a significant source of gaseous and
vapour-phase pollutants, such as $SO_2$, CO and $NO_x$ (Masiol and Harrison, 2014). In the same vein, mobile sources,
such as vehicle exhaust, generally contribute to the increase in CO and $NO_x$ levels, as motor vehicle emissions
are the dominant sources of CO and $NO_x$ emissions in urban areas (Yu et al., 2004). Given that Barajas airport is
situated near Madrid and significantly influenced by external sources, particularly traffic on the southwest side of
the airport, it experiences considerable environmental impact. Therefore, the ratios of $SO_2/NO_x$ and CO/$e$BC were
used in this analysis as indicators of the relative emission strengths associated with aircraft movements. The
$SO_2/NO_x$ ratio would increase in the case of aviation emissions compared to traffic emissions, since $NO_x$ emissions
from aircraft are difficult to distinguish due to the major influence of other sources (Yu et al., 2004; Carslaw et
al., 2006). Consequently, in situations where there are substantial levels of $NO_x$ emissions, the $SO_2/NO_x$ ratio will
be low due to the impact of on-road vehicles emissions. This enables the identification of aircraft's relative
contribution at the airport, as shown in Fig.9. The analysis of the $SO_2/NO_x$ and CO/$e$BC concentration ratios at
Madrid-Barajas Airport in October 2021varies based on wind direction and speed. The bivariate polar plots shown
in Fig. 9 indicate higher $SO_2/NO_x$ and CO/$e$BC ratios were measured when dominant winds originating from the
northeast of the airport, where there was minimal or no contribution from road traffic. The higher $SO_2/NO_x$ and
CO/$e$BC ratios suggest the potential impact of aircraft taxing and taking off on local ambient $SO_2$ and CO
concentrations, particularly when winds originate from northeast, where the 18L/36R runways are located. $SO_2$
emissions are primarily associated with the sulphur content of the fuel and emissions from aircraft activities at the
airport, such as approach, taxi-idle and climb. As a result, $SO_2$ plays a significant role in tracing aircraft emissions
at a local scale (Yang et al., 2018). Black carbon ($e$BC) and carbon monoxide (CO) are primarily produced by
incomplete or inefficient combustion. Around the airport perimeter, aircraft are a significant contributor to CO
emissions. Therefore, it's possible for aircraft engines to emit more CO compared to emissions from road traffic,
due to the duration spent at the airport in taxiing /idling mode (Yu et al., 2004; Zhu et al., 2011). The CO/$e$BC
ratio significantly varies with the source (Bond et al., 2004), indicating the presence of different emission sources
in the vicinity of the airport, as previously reported. The highest levels of CO from aircraft are emitted at low
engine power settings, such as during taxiing and idling. This significantly impacts air quality within the airport
perimeter, as idle and taxi phases constitute the majority of the time an aircraft spends at the airport (Stettler et
al., 2011; Yunos et al., 2017). Higher CO/$e$BC ratio in air parcels originating from the northeast can also be
attributed to aircraft activity on runways 18L/36R, which is located northeast of the measurement station.
Conversely, $SO_2$ /$NO_x$ and CO/$e$BC ratios were lower (ranging from 0 to 0.4) when winds originated from the
southwest, due to significant sources of $NO_x$ and $e$BC in this direction, such as nearby road traffic. Based on the
polar plots shown in Fig. 9, an aircraft $SO_2$ and CO signal is identified to the east and northeast, distinct from the
wind-dependant $NO_x$ pattern. Further details regarding the daily variation of meteorological parameters and trace
gases during the sampling period are available in the supplementary material (Fig. S1).


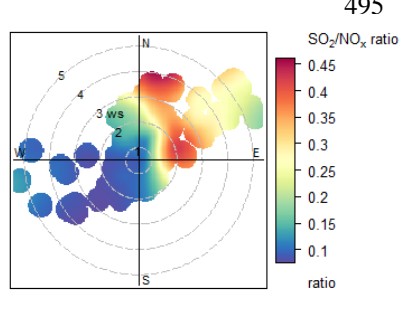
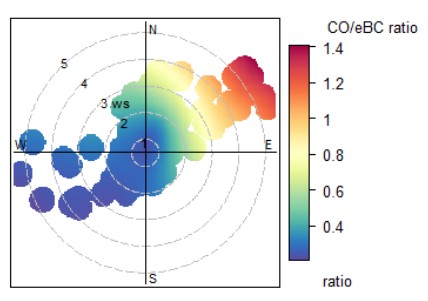

**Figure 9. Bivariate polar plots of $SO_2$ /$NO_x$ and CO/$e$BC ratios at the airport. The angular contributions of $SO_2$ and**
**CO is different compared to the PMF determined factors. The plots indicates that the flight activities at the east and**
**northeast where the 18L/36R runway is located are the source of increase in $SO_2$ and CO.**
**4. Conclusion**
This study identified the impact of an international airport on the local air quality. As part of the AVIATOR
campaign, several measurements were conducted at the Madrid–Barajas Airport, in October 2021 for monitoring
the chemical composition of sub-micron particles and ambient trace gas concentrations near runway. Assessing
the impact of Madrid–Barajas Airport emissions on local air quality is challenging because of the complex nature
of airport emissions and the strong influence from urban emissions. The proximity of the airport to urban areas,
major highways, roads, and terminal buildings (T1, T2, T3, T4 and TS4) further complicates the task, making it
difficult to clearly identify the specific contributions of aircraft emissions. However, aircraft emissions are
characterized by high levels of unburned hydrocarbons, $SO_2$, CO and particulate black carbon ($e$BC) which are
more concentrated around the airport facilities and runways. Therefore, looking at elevated levels of these markers
might indicate a stronger influence from aviation-related activities, especially during times of high airport traffic.
Total non-refractory particles were dominated by organics (more than 72% of the total). Sulphate particles were
the second most abundant chemical species and accounted for about 13% of the total aerosol. Based on AMS data
(Ratio of $m/z$ 85:71), no significant oil fraction in the organic particulate matter (PM) samples were measured.
This could indicate the absence of oil in sub-micron particle size range or due to the method used in this study
(AMS) is not able to identify lubricant oil in PM. Thus, further measurements with improved measurement
technique may be required to identify oil fraction in sub-micron organic aerosol.  Trace gases were also monitored
along with the particle monitoring tools. Average ambient concentrations of $e$BC, $NO_x$, $SO_2$, $PM_{2.5}$, $PM_{10}$ at the
airport during October 2021 were 1.07, 22.7, 4.10, 9.35, and 16.43 (μg/m³), respectively. $NO_x$ contribution at the
sampling point was highest when the winds originating from south and southeast of the airport. There are two
motorways with road traffic are located at the same direction as well as terminal buildings and southern runways.
Therefore, $NO_x$ concentrations were more likely determined by on-road traffic compared to the aircraft activity at
the sampling point. Sources of organic aerosols (as the most abundant non-refractory aerosol group) were
identified using Positive Matrix Factorisation (PMF) analysis. PMF was able to discriminate three main significant
sources: Less Oxidised Oxygenated Organic Aerosol (LO-OOA), Alkane Organic Aerosol (AlkOA), and More
Oxidised Oxygenated Organic Aerosol (MO OOA). The sum of LO-OOA and MO OOA fractions accounting for
more than 80% of the total organic mass throughout the campaign, LO-OOA had the highest relative intensity
(RI) at *m/z* 43 (which is characteristic of LO-OOA), MO-OOA had a high RI at *m/z* 28 and 44 these indicate a
potential secondary aerosol fraction. Third factor, AlkOA, had high RIs at *m/z* 43, 57 and 85 (attributed to decane
previously) which is related to jet fuel vapour (Smith et al., 2022). Bivariate polar plots were used to angular PMF
determined factor and ambient trace gas distributions based on wind speed and wind direction at the airport. It has
been found that, the PMF determined factors had highest relative contributions when the winds originating from
the west and southwest of the airport where runways 14R/32L and 18R/36L, as well as terminals T1, T2, T3, T4
and TS4, are located. The $SO_2/NO_x$ and CO/*e*BC ratio have been shown to represent a useful tool for assessing
relative emission strength associated with aircraft movements. Take-off activities at the northeast of the
measurement station were identified as a potential local source of $SO_2$ and CO in Barajas-Madrid. Angular
correlation analysis based on wind direction and speed indicated that *e*BC and THC emissions are potentially
determined by aircraft take off activities at 18L/36R runway located along the east and northeast of the sampling
point where more than 50% of the take-off activity took place in the sampling period.
There are two previously reported significant ways to reduce aviation emissions at airports, improving efficiency
of the processes emitting air pollutants such as electrification of airport taxiway operations (Salihu et al., 2021),
and switching to sustainable alternative fuels where applicable. Improved ground activities at airports such as
electric aircraft towing system can potentially lead up to 82 % reduction in $CO_2$ emissions (van Baaren, 2019),
while switching to SAF alone reduce Landing-takeoff cycle (LTO) emissions up to 70 % compared to fossil fuel
(Schripp et al., 2022). Further, SAF use for auxiliary power units (APU) also potentially reduce $NO_x$ and $CO_2$
emissions by at least 5%. Therefore, improving energy efficiency of ground activities at airports and using SAF
are recommended for policymakers to improve the overall air quality at airports.
*Author contributions*. **Saleh Alzahrani, Doğuşhan Kılıç, Michael Flynn, Paul I. Williams** and **James Allan**
designed the project; **Saleh Alzahrani, Doğuşhan Kılıç, Michael Flynn** and **Paul I. Williams** performed the
fieldwork; **Saleh Alzahrani** performed the data analysis, and wrote – original draft of the article; **Doğuşhan
Kılıç** reviewed and edited the article; **Paul I. Williams** and **James Allan** supervised, reviewed and edited the
article.
*Competing interests.* At least one of the (co-) authors is a member of the editorial board of Atmospheric
Chemistry and Physics.
**Acknowledgments**
This project has received funding from the European Union's Horizon 2020 research and innovation programme
under Grant Agreement No 814801.

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
