# Peer review of "International airport emissions and their impact on local air quality: Chemical speciation of ambient aerosols at Madrid Barajas Airport during AVIATOR Campaign"

_EGUsphere, 2023_

## Author Comment (AC1)

[Figure]

**Figure S3. The diurnal pattern of PM2.5 concentrations, measured in micrograms per cubic meter, shows variations between a sampling point and the southern area of the airport, which are approximately 6km apart during autumn campaign (WP4) of AVIATOR.**

[Figure]

**Figure S2. Total number of flights per hour counted at Madrid–Barajas airport during autumn campaign (WP4) of AVIATOR**

[Figure]

**Figure S4. Bivariate polar plots of lubrication oil ratio measured by AMS ratios at the Madrid-Barajas International Airport during autumn campaign (WP4) of AVIATOR.**

(a)                                                                          (b)

[Figure]

**Figure S5. Calendar plots for lubrication oil ratio with annotations highlighting those days where the ratio of lubrication oil > 0.66 (a), Hour of the day analysis for the lubrication oil ratio during autumn campaign (WP4) of AVIATOR (b)**

---

## Author Comment (AC2)

[Figure]

**Figure S3. The diurnal pattern of PM2.5 concentrations, measured in micrograms per cubic meter, shows variations between a sampling point and the southern area of the airport, which are approximately 6km apart during autumn campaign (WP4) of AVIATOR.**

(a)

(b)

[Figure]

**Figure S5. Calendar plots for lubrication oil ratio with annotations highlighting those days where the ratio of lubrication oil > 0.66 (a), Hour of the day analysis for the lubrication oil ratio during autumn campaign (WP4) of AVIATOR (b)**

[Figure]

**Figure S4. Bivariate polar plots of lubrication oil ratio measured by AMS ratios at the Madrid-Barajas International Airport during autumn campaign (WP4) of AVIATOR.**

[Figure]

**Figure 3. Temporal variability of lubrication oil fraction in total aerosol mass obtained from AMS measurements.**

[Figure]

**Figure S2. Total number of flights per hour counted at Madrid–Barajas airport during autumn campaign (WP4) of AVIATOR**

---

## Author Response (AR1)

Thanks to the reviewer for their thorough examination of the manuscript and their valuable feedback. We have carefully considered all the comments provided and have incorporated the suggested improvements into the updated version of the manuscript.

**RC1**

1- With mention of source apportionment, more attention should be paid to understanding the urban contributions. An attempt to quantify the relative contributions of urban vs aviation would be welcome but, if not possible, an explanation of what would be needed to do so would be needed.

Response: Thank you for pointing this. We provided background particle matter concentrations PM2.5 in comparison with PM2.5 at sampling point. We have added these lines to discuss the potential background impact as suggested by the referee comment.

This is supported by the variance in background particulate matter concentrations located south of the airport compared to those at the sampling point, approximately 6 km apart, as illustrated in Fig. S3. (Fig. S3) reveals that PM2.5 levels at both locations experience significant increases during morning and evening rush hours, with the sampling point consistently showing higher concentrations than the background location. The diurnal pattern of the background location demonstrates a rapid decrease in PM2.5 levels in the afternoon, unlike the measurements at the sampling point. Additionally, there is a noticeable lag of about an hour between the peak concentrations at the sampling point and those in the background, suggesting the influence of additional combustion sources of PM2.5, notably aviation-related activities, particularly during periods of increased airport traffic.

[Figure]

2- an illustrative figure of a time series of 1) an aircraft take-off and 2) an aircraft landing would provide insight into the detailed contribution of aviation compared to the more diffuse background contributions.

Response: Diurnal flight activity been provided to be compared with PM2.5 concentrations located south of the airport and at the sampling point as well as the AlkOA factor.

[Figure]

3- Terminology: Use of SV-OOA—this should be rephrased to "less oxidized oxidized organic aerosol" (LO-OOA) as there is no specific volatility data presented (such a thermal denuder) to determine if it is semi volatile or more volatile than MO-OOA.

Response: Thank you for pointing this out. SV-OOA been rephrased to LO-OOA and we have added these lines for clarity.

Another study by Smith et al. (2022), investigated the chemical composition of organic aerosols emitted by gas turbines and identified a Semi-Volatile Oxygenated Organic Aerosol (SV-OOA) factor, which forms through oxidative processes near the engine exit. A strong correlation (R = 0.91) and similarity in mass spectra between the LO-OOA in this study and the SV-OOA described by Smith et al. (2022) were observed. Owing to the absence of volatility measurements during this period and the limited time for aging (no more than a few minutes), we consider the LO-OOA factor in our analysis to be the most accurate estimate available, rather than the SV-OOA as suggested by Smith et al. (2022).

4- Line 59-60: Jet A1 (in Europe and Jet A in US) is a mix of BOTH aliphatic HCs and **aromatic species** (not just aliphatics). A minimum aromatic content is prescribed, in fact, which is relevant as sustainable fuels are considered. So the aromatic content cannot be ignored.

Response:  Thank you for the comment. We rephrased the sentence based on referee comment. We have added these lines about the aromatic fraction.

 Aviation fuel Jet A1 is the most common type of fuel that is used in civil aviation. It's a complex mixture of aliphatic hydrocarbons and aromatic compounds, characterized by a mean C/H ratio of ~ 0.52 (with an average empirical molecular formula of C12H23) (Lee et al., 2010). The paraffins fractions in jet fuel typically make up over 75% of the fuel by weight, while the aromatic content is less than or equal to 25% (Liu et al., 2013).

5- Line 91: "is the main international airport" , perhaps add "in Spain"? Otherwise it could be interpreted more broadly.

Response:  Thank you for the suggestion. We rephrased it as Adolfo Suárez Madrid-Barajas Airport is the main international airport in Spain.

6- Section 2.3 **Data Analysis**: This section is poorly written and needs to be edited.  The verb tense changes throughout ("was operating" vs "were analyzed", etc.) There is significant redundancy and 149-151 is an incomplete sentence or missing verb, etc. Line 148 should read SQUIRREL "operated within Igor Pro", not "supplied by Igor Pro".

Response: Thank you for your comment. We improved the writing of this section based on the referee comments.

7- Regarding the apparent lack of observation of oil emissions (lines187-202), it might be valuable to cite Ungeheuer et al., 2021 and Fushimi 2019 in line 187

Response: Thank you for suggesting these references to include. They have been added to the manuscript.

8- Some insight on this might be gained by coloring the data stream in figure 3 with the wind direction. If the "little oil" region can be attributed to runway or terminal influenced air and the "unlikely" region is more attributable to urban background, it may well be that the oil (and aviation signal more generally) is swamped by urban background.

Response: Thank you for your feedback. Bivariate polar plots of lubrication oil ratio measured by AMS is provided to describe the oil contribution. These lines been added to the manuscript for more clarity.

Additional information on how the lubrication oil ratio, as measured by AMS, varies with wind speed and direction, is provided in the supplementary material (Fig.S4). During the AVIATOR autumn campaign, measuring oil was challenging due to the prevalent urban background. A "little oil" region was identified at low to moderate wind speeds (2~5 m/s) originating from the southwest, encompassing terminal buildings (T1, T2, T3, T4, and TS4), two runways (14R/32L and 18R/36L), and a hangar zone. In contrast, a region "unlikely to contain oil" was noted when winds came from the northeast of the airport, near runways 18L/36R, with relatively higher wind speeds (above 5 m/s).

[Figure]

Moreover, Calendar plots for lubrication oil ratio with annotations highlighting the days where the ratio of lubrication oil > 0.66 are also provided. These lines been added to the manuscript for more clarity.

Fig.S5 displays the daily lubrication oil ratio throughout the sampling period, pinpointing Sunday, October 16th, as the only day when the lubrication oil ratio surpassed 0.66. On other days, the ratio suggested a minimal likelihood of oil presence. An hourly analysis within Fig.S5 reveals that the lubrication oil ratio exceeded 0.66 only at 20:00, aligning with the evening peak in PM2.5 concentrations Fig.S3. This suggests a significant influence of urban background aerosols on the lubrication oil measurements.

[Figure]

9-    the last sentence (line 202) might better say that the "organic mass may be under-represented in this study".

Response: Thank you for the suggestion. sentence been rephrased.

10-    line 227: "an earlier studies" singular/plural mismatch.

Response: Typo been corrected.

11- line 248 "Aliphatic#1" is unclear. If this is referencing PMF analysis from a separate paper, this needs to be rephrased and referenced. (vs AlkOA already described in this manuscript).

Response:  Thank you for your comment. This been adjusted to primary aliphatic factor and referenced in the manuscript.

12- line 287-288 discussion of SV-OOA and MO-OOA formation: might be worth emphasizing here that this processing occurs primarily for urban contributions in this study since there is too little time for significant photochemical oxidation of aviation emissions so close to the source.

Response: Thank you for your feedback. This been mentioned in the manuscript and these lines been added for more clarity.

In this study, urban contributions are predominantly subject to this processing, as there is insufficient time for significant photochemical oxidation of aviation emissions in such close proximity to the source.

**RC2**

1- The discussion overall lacks consideration of the effect of background urban pollution on the measurements performed in this work. Particularly, given the diurnal trend of the pollutants presented in Figures 5 and 6, it is evident that many pollutants show a strong influence from urban emissions. This strong influence from urban emissions could potentially alter some of the main conclusions in the current manuscript.

Response: Thank you for bringing this to our attention. We have included information on background particle matter concentrations (PM2.5) and compared them with the PM2.5 levels at the sampling point. These additions address the potential background impact, as suggested by the referee's comments.

This is supported by the variance in background particulate matter concentrations located south of the airport compared to those at the sampling point, approximately 6 km apart, as illustrated in Fig. S3. (Fig. S3) reveals that PM2.5 levels at both locations experience significant increases during morning and evening rush hours, with the sampling point consistently showing higher concentrations than the background location. The diurnal pattern of the background location demonstrates a rapid decrease in PM2.5 levels in the afternoon, unlike the measurements at the sampling point. Additionally, there is a noticeable lag of about an hour between the peak concentrations at the sampling point and those in the background, suggesting the influence of additional combustion sources of PM2.5, notably aviation-related activities, particularly during periods of increased airport traffic.

[Figure]

2- Section 2.1: Please note the distance from sampling location to the runways, terminals, and nearest highway in Fig.1 or text.

Response: Thank you for your comment. This been addressed in the following lines.

The distance from sampling location to the runways 18L/36R, 18R/36L, 14L/32R and 14R/32L are 680 m, 620 m, 3.2 km, and 4.1 km respectively. Furthermore, the distance between sampling location and adjacent terminals T1, T2, T3 is approximately 5 km whereas 3 km and 1.5 km to the terminals T4 and TS4 respectively. Nearest highway is located around 2.6 km away from the sampling location.

3- Section 2.2: Please give a more detailed description of AMS, including the AMS inlet used here (PM1 or PM2.5?), and the time resolution of AMS.

Response: Thank you for the suggestion. More detailed description of AMS been included to the manuscript.

An aerodynamic lens is used to draw aerosols into a vacuum chamber. Particles are focused into a narrow beam and accelerated to a velocity inversely related to their vacuum aerodynamic diameter. The particles impact on a tungsten surface, heated to 600 °C, which causes them to flash vaporise. A 70-eV electron is used to ionize the vapours before they are analysed by mass spectrometry. During the measurement period, AMS was sampling with 1μm cut-off inlet and at 30 s time resolution.

4- Section 2.3: Please add more details of how the authors selected the PMF factors and explain the technical terms used (e.g. FPEAK, Q/Qexp).

Response: These lines been added to the manuscript for more clarity

Seed runs and fpeak are rotational techniques in ME-2 tool and they are one of the unconstrained PMF run approaches used for the exploration of the solution space.

Furthermore, the obtained solution exhibited the most favorable results, characterized by distinct diurnal trends and dissimilarities in time series and mass-to-charge ratios among the factors.

5- Rewrite sentence line 149-151.

Response: sentence been rewritten and added to the manuscript

The Source Finder (SoFi) is a software package designed to analyse multivariate data using state-of-the-art source apportionment techniques to understand the sources of various pollutants (Canonaco et al., 2013)

6- Section 3.1: are the signal at m/z 85 and m/z 71 used here all from AMS UMR analysis?

Response: Yes, signal at m/z 85 and m/z 71 from AMS UMR. As suggested by (Yu et al., 2012) the m/z 85 signal is an oil signal in the AMS mass spectrum. The ratio of m/z 85:71 is used as a marker for oil. The value of 0.66 is used as a benchmark for oil lubrication identification.

7- When comparing the results from this study to those presented by Yu et al. in 2012, it's important to consider other factors that may be contributing to the very low presence of oil-related aerosols in your findings. These factors could include the overall mass loading of aerosols, the influence of urban aerosol emissions, or the proximity of the sampling location to the nearest runways.

Response: Thank you for your suggestions. This been addressed as follow:

Additional factors that could potentially impact the minimal presence of oil lubrication in this analysis might involve the overall mass loading of aerosols, the influence of urban aerosol emissions, or the proximity of the sampling point to the nearest runways.

Calendar plots for lubrication oil ratio with annotations highlighting the days where the ratio of lubrication oil > 0.66 are also provided. These lines have been included in the manuscript to enhance clarity.

Calendar plots displays the daily lubrication oil ratio throughout the sampling period, pinpointing Sunday, October 16th, as the only day when the lubrication oil ratio surpassed 0.66. On other days, the ratio suggested a minimal likelihood of oil presence. An hourly analysis within Calendar plots

reveal that the lubrication oil ratio exceeded 0.66 only at 20:00, aligning with the evening peak in PM2.5 concentrations Fig.S3. This suggests a significant influence of urban background aerosols on the lubrication oil measurements.

[Figure]

Moreover, Bivariate polar plots of lubrication oil ratio measured by AMS is provided to describe the oil contribution. These lines been added to the manuscript for more clarity.

Additional information on how the lubrication oil ratio, as measured by AMS, varies with wind speed and direction, is provided in the supplementary material (Fig.S4). During the AVIATOR autumn campaign, measuring oil was challenging due to the prevalent urban background. A "little oil" region was identified at low to moderate wind speeds (2~5 m/s) originating from the southwest, encompassing terminal buildings (T1, T2, T3, T4, and TS4), two runways (14R/32L and 18R/36L), and a hangar zone. In contrast, a region "unlikely to contain oil" was noted when winds came from the northeast of the airport, near runways 18L/36R, with relatively higher wind speeds (above 5 m/s).

[Figure]

8- The values displayed in Fig 3 are noisy and show a broad range. It may be beneficial to further average or smooth these values for a clearer presentation of the data.

Response: Thank you for your comment. Oil marker plot has been adjusted accordingly.

[Figure]

9- Section 3.2: the main conclusions in this part are same as previous studies. Are there any new findings from this work?

Response: Thank you for your feedback. Yes, there are. We measured and identified indene as a marker at Madrid Airport in the near runway ambient samples. The following lines been added to discuss the findings.

The first factor in Fig.4, LO-OOA, stands for Less Oxidised Oxygenated Organic Aerosol. It is a type of secondary organic aerosol (SOA) characterized by its low degree of oxidation. LO-OOA are formed in the atmosphere through the oxidation of volatile organic compounds (VOCs), which can originate from a variety of anthropogenic sources. In this analysis LO-OOA shows the presence of an aromatic marker at m/z 115, a marker used for identifying indene (C9H8) ion in previous studies focusing on aviation emissions (Timko et al., 2014; Smith et al., 2022). LO-OOA is associated with aromatic fragments at m/z 77, 91, 105, 115 and presents a high relative intensity (0.13) at m/z 43

(characteristic of LO-OOA) and a lower relative intensity (<0.04) at m/z 91 which is related to toluene ion (C7H7) (Smith et al., 2022). Ambient temperature plays a crucial role in influencing the LO-OOA factor, displaying significant diurnal fluctuations. The lowest concentrations of LO-OOA are recorded at midday, coinciding with the peak in ambient temperatures (Fig. 5).

10- The sources of MOOOA need to be discussed.

Response: Thank you for your comment. A more detailed discussion on MOOOA been added to the manuscript.

The third factor, More Oxidised Oxygenated Organic Aerosol (MO-OOA), is a type of secondary organic aerosol (SOA) that can form from various origins and processes, such as photochemical processing of aged SOA and the regional-scale transport of chemical reactions. MO-OOA has a spectral fingerprint that consists of more oxidised ions (compared to LO-OOA and AlkOA), indicating a secondary aerosol fraction in the sample. MO-OOA is characterized by its notably high relative intensities (>0.18) at m/z 29 and 44, which serve as markers for its identification. Given that MO-OOA has the highest f44/43 ratio among the three factors, it is expected to be the most oxygenated (in terms of chemical content) factor. Being more oxidised potentially makes MO-OOA less volatile than LO-OOA (Jimenez et al., 2009; Smith et al., 2022). MO-OOA in this analysis indicates the formation of aged secondary organic aerosols with no significant diurnal variation (Fig. 5), often associated with air masses transported from polluted regions.

11- Line 248: the term "aliphatic #1" is from the reference, please avoid using it directly without explanation.

Response: Thank you for your comment. This been adjusted to primary aliphatic factor and referenced in the manuscript.

12- Section 3.3: concentration of AlkOA is suggested to be influenced by flight activity, please include such information in main text or SI.

Response: Thank you for your comment. Diurnal flight activity data has been provided for comparison with the AlkOA factor. These lines been added to the manuscript

The concentration of AlkOA factor is relatively higher in the afternoon compared to the morning and midday. The pattern of diurnal AlkOA closely resembles that of diurnal flight activities, suggesting that the surge in AlkOA levels beginning at noon is linked to primary particles released by aircraft. Further details on daily aircraft activities can be found in the supplementary material (Fig. S2)

[Figure]

13- Line 291: please further explain the method of normalization.

Response: Thank you for your comment. This been explained as the following.

Normalising is accomplished by dividing the concentrations of the pollutants by their average value.

14- Line 298-300: the authors suggested the diurnal trend of the pollutants listed in Fig 6 were all similar, but it is very clear that BC, NOx, and CO have two peaks, total number concentration starts to increase mid-morning and maintains high during the daytime, while AlkOA only has one big peak around evening. Please provide further evidence for the conclusion. In Fig 6, at hour 23, the normalized level of AlkOA is about 1.9, but at hour 0, it's about 0.8, can you please explain the big change here? I'm assuming the change from 1.9 to 0.8 happened within one hour since this plot describes diurnal cycle.

Response: Thank you for your comment. This has been addressed as below:

The daily trend of eBC, NOx and CO are mostly similar, with very pronounced increases in concentrations during the morning and evening rush hours. The average concentrations were 1.07 µg/m3, 22.7 µg/m3 and 0.23 mg/m3 for eBC, NOx and CO respectively (Table S1). AlkOA gradually increases during the morning, with multiple minor peaks observed in the morning hours. The average concentration of AlkOA is higher at night than during the day. This increase is potentially related to daily aircraft activities. AlkOA began to increase, reaching a maximum during the afternoon rush hour from 12:00-18:00. a second rapid increase occurred around 20:00, potentially caused by an increase in the number of flights at this time (Fig. S2). Early morning AlkOA concentrations are significantly lower compared to those of eBC, NOx and CO. This difference could be attributed to reduced emissions resulting from decreased aircraft activities in early mornings (Fig. S2). The rise in trace gases and eBC observed in the early morning hours could originate from various airport operations. Such operations might encompass emissions from auxiliary power units, vehicle traffic, and the use of ground service equipment at the airport (Masiol and Harrison, 2014). The total number concentration exhibited a temporal pattern similar to that of AlkOA from 15:00–21:00. Likewise, the temporal profiles of AlkOA and trace gases were similar during the afternoon period (17:00-21:00). This similarity in temporal profiles suggests common source origins, which may be temporally associated with aircraft activity or the influence of background urban pollution.

15- Section 3.4 Fig.8 and Fig.9: Please explain why the authors selected AlkOA vs. eBC and SVOOA vs. THC for correlation analysis.

Response: Thank you for your comment. This been addressed as follow:

- Based on the strength of the relationship outlined in Table 1 between derived factors and external tracers, the linear correlations (Pearson correlation) between (i) AlkOA with eBC and (ii) LO-OOA with THC

- Both AlkOA and eBC are related to jet fuel emissions, as they are directly emitted by aircraft engines as a result of fuel combustion. eBC emissions are a function of engine power settings, reaching their maximum at full thrust during take-off (Kinsey et al. 2011; Hu et al., 2009).

- THC emissions at airports primarily dependent on the jet engine thrust setting (Anderson et al., 2006; Onasch et al., 2009). When engines operate at low thrust settings (e.g., during landing, taxiing, idling), combustion is less efficient, leading to the emission of higher amounts of hydrocarbons. The association between LO-OOA and THC in certain areas of the airport can be interpreted as indicative of fresh emissions from aircraft in service.

16- why selected SO2/NOx and CO/eBC ratios for analysis of aircraft activity? What do these ratios imply in terms of aircraft emission?

Response: Thank you for your comment. This has been addressed and included in the manuscript as follows:

Given that Barajas airport is situated near Madrid and significantly influenced by external sources, particularly traffic on the southwest side of the airport, it experiences considerable environmental impact. Therefore, the ratios of SO2/NOx and CO/eBC were used in this analysis as indicators of the relative emission strengths associated with aircraft movements.

The SO2/NOx ratio would increase in the case of aviation emissions compared to traffic emissions, since NOx emissions from aircraft are difficult to distinguish due to the major influence of other sources (Yu et al., 2004; Carslaw et al., 2006). Consequently, in situations where there are substantial levels of NOx emissions, the SO2/NOx ratio will be low due to the impact of on-road vehicles emissions. This enables the identification of aircraft's relative contribution at the airport, as shown in Fig.9.

SO2 emissions are primarily associated with the sulphur content of the fuel and emissions from aircraft activities at the airport, such as approach, taxi-idle and climb. As a result, SO2 plays a significant role in tracing aircraft emissions at a local scale (Yang et al., 2018).

Black carbon (eBC) and carbon monoxide (CO) are primarily produced by incomplete or inefficient combustion. Around the airport perimeter, aircraft are a significant contributor to CO emissions. Therefore, it's possible for aircraft engines to emit more CO compared to emissions from road traffic, due to the duration spent at the airport in taxiing /idling mode (Yu et al., 2004; Zhu et al., 2011).

The CO/eBC ratio significantly varies with the source (Bond et al., 2004), indicating the presence of different emission sources in the vicinity of the airport, as previously reported.

17- The authors suggested CO was mainly related to road traffic emissions based on previous paper (line 379-380), then the authors related CO measured by the monitoring site with aircraft activities (line 390-392). Could you further explain your discussion and conclusion here?

Response: Thank you for your comment. Being predominantly emitted by on-road vehicles, at low engine settings (taxi/idle) aircraft also emit significant amount of CO. This has been clarified in the manuscript as follows:

Aviation activities have previously been reported as a significant source of gaseous and vapour-phase pollutants, such as SO2, CO and NOx (Masiol and Harrison, 2014). In the same vein, mobile sources, such as vehicle exhaust, generally contribute to the increase in CO and NOx levels, as motor vehicle emissions are the dominant sources of CO and NOx emissions in urban areas (Yu et al., 2004). Around the airport perimeter, aircraft are a significant contributor to CO emissions. Therefore, it's possible for aircraft engines to emit more CO compared to emissions from road traffic, due to the duration spent at the airport in taxiing /idling mode (Yu et al., 2004; Zhu et al., 2011).

---

## Referee Report (RR1)

This paper presents an investigation into the impacts of international airport emissions on local air quality, providing insights into the chemical speciation of ambient aerosols at Madrid-Barajas Airport. The authors have addressed the reviewers' comments comprehensively, including additional data, clarifying methodologies, and enhancing the discussion on urban pollution's influence. Although minor revisions are needed to further refine the grammar and improve the clarity of some sections, the overall quality of the research is high. The findings contribute significantly to the understanding of aviation emissions and their effects on air quality, making this study a valuable addition to the field.

1. I strongly advise the authors to carefully check their manuscript or seek assistance from proofreading services to ensure it does not contain any typos, grammar errors, awkward phrasing, formatting errors, or any other issues that could cause misunderstandings. For example, lines 193-196 need to be rephrased to avoid repeating information. It should also be noted that the nitrate and sulphate species measured by AMS could contain organic nitrogen or organic sulfur. There is a typo in line 197, and the sentence in lines 199-200 is incomplete. While I will not list all the sentences that need correction here, I recommend that the authors carefully read and thoroughly check their manuscript.

2. Please add the AMS ion chemical formula when discussing the ions at certain m/z.

3. Provide a more detailed description of the two different types of lines in Figure 3.

4. Include the standard deviations of concentrations in Figure 5.

5. Lines 305 and 341-345: The authors suggest that AlkOA concentration is associated with aircraft activities because its concentration starts to increase at noon. However, it is evident in Figure S2 that the total number of flights is higher between 8:00 am and 2:00 pm compared to other times of the day, which is not consistent with the AlkOA concentration described in Figure 5. Please provide further details to support the authors' statement and conclusion in this section.

6. Section 3.3: For the correlation analysis of the three PMF factors with other studied pollutants, is the analysis done using the data collected from the whole campaign or the average data from the diurnal variation analysis? If the authors used the one-hour average data from the diurnal variation analysis, I suggest rerunning the analysis using the data from the whole campaign.

7. Lines 332-333: The authors suggest that the concentrations of BC, NOx, SO2, and CO could be influenced by meteorological conditions, which might explain

the moderate correlation observed between AlkOA and these pollutants. However, given that AlkOA is measured as part of AMS sub-micron samples, it is reasonable to assume that AlkOA would also be influenced by meteorological factors. Additionally, since BC is also in the particle phase, the influence of meteorological conditions on both AlkOA and BC should be similar.

8. Figures 5 and 6: The authors did not address the previous question about the significant decrease in AlkOA from hour 23 to hour 0, which could be associated with the large standard deviation when averaging the data to study the diurnal variation. I recommend that the authors consider the standard deviation when discussing the diurnal variation if it is large.

---

## Author Response (AR2)

We thank the reviewers for their thorough examination of the manuscript and their valuable feedback. We have carefully considered all the comments provided and have incorporated the suggested improvements into the updated version of the manuscript.

**Reviewer 2**

1. *lines 193-196 need to be rephrased to avoid repeating information. It should also be noted that the nitrate and sulphate species measured by AMS could contain organic nitrogen or organic sulfur. There is a typo in line 197, and the sentence in lines 199-200 is incomplete.*

Response: Thank you for your comment. We improved the writing based on the reviewer comments.

- We rephrased the sentence and have added these lines about the nitrate and sulphate species

The average mass concentration of organic and inorganic aerosols during the entire campaign was 9.6 μg/m³. The bar chart in Fig. 2 shows aerosol fractions, with organic species accounting for more than 70% of the total aerosols. This is significantly higher than the nearest component, sulphate, which accounted for 15%. It should also be noted that the nitrate and sulphate species measured by AMS could potentially contain an organic fraction.

- We corrected the typo
- We rephrased the incomplete sentence

The PMF analysis in this paper primarily focuses on the composition of the organic mass concentration, which is discussed in further detail in Section 3.2

2. *Please add the AMS ion chemical formula when discussing the ions at certain m/z.*

Response: Thank you for pointing this out. We have added the ion chemical formula as suggested by the reviewer and included a reference to the manuscript.

The m/z 85 signal is a well-known oil marker in the AMS mass spectrum, attributed to synthetic esters ($C_5H_9O^+$)

LO-OOA is associated with aromatic fragments at m/z 77 ($C_6H_5^+$), and 105 ($C_8H_9^+$). It presents a high relative intensity (0.13) at m/z 43 ($C_3H_7^+$) (characteristic of LO-OOA) and a lower relative intensity (<0.04) at m/z 91, which is related to toluene ion ($C_7H_7^+$) (Timko et al., 2014; Smith et al., 2022).

MO-OOA is characterized by its notably high relative intensities (>0.18) at m/z 29 ($CHO^+$) and 44 ($CO_2^+$), which serve as markers for its identification (Alfarra et al., 2007)

3. *Provide a more detailed description of the two different types of lines in Figure 3.*

Response: Thank you for your comment. Figure 3 description is edited to provide required details according to comments:

A smooth red line is fitted to the data, while the dashed black line represents the value of 0.66, assumed for oil-free organic PM emitted from aircraft engines. The analysis showed that no oil or very little (<5%) oil fraction was detected.

4. *Include the standard deviations of concentrations in Figure 5.*

Response: Thank you for your comment. Statistical details including sd of the obtained factors already tabled in the supplementary material Table S1. We have added this line for clarity and updated Figure 5 along with its caption according to the comments:

Detailed statistics of the obtained factors for the entire campaign are provided in the supplementary material (Table S1)

[Figure]

**Figure 5. Diurnal pattern of the solved factors from October 8, 2021, to October 23, 2021. The mean diurnal pattern is shown as solid lines, and the shading indicates the 95% confidence interval for the mean.**

5. *Lines 305 and 341-345: The authors suggest that AlkOA concentration is associated with aircraft activities because its concentration starts to increase at noon. However, it is evident in Figure S2 that the total number of flights is higher between 8:00 am and 2:00 pm compared to other times of the day, which is not consistent with the AlkOA concentration described in Figure 5. Please provide further details to support the authors' statement and conclusion in this section.*

Response: Thank you for your comment. The AlkOA factor increases between 09:00 and 20:00 with the rise in flight activity. The increase in AlkOA between 22:00 and 23:00 is not statistically significant due to high variability. The discussion of the results has been updated as follows:

The AlkOA factor shows an increase between 09:00 and 20:00 and again between 22:00 and 23:00. Based on the mean diurnal pattern with a 95% confidence interval, the AlkOA factor increases during the 09:00 to 20:00 period, corresponding with peak flight activity (approximately 71% of total flights). Further details on daily aircraft activities are provided in the supplementary material (Fig. S2). The increase in AlkOA between 22:00 and 23:00 is not statistically significant due to high variability (Fig. 5).

6. *Section 3.3: For the correlation analysis of the three PMF factors with other studied pollutants, is the analysis done using the data collected from the whole campaign or the average data from the diurnal variation analysis? If the authors used the one-hour average data from the diurnal variation analysis, I suggest rerunning the analysis using the data from the whole campaign.*

Response: Thank you for your comment. The analysis conducted using data collected throughout the entire campaign. we have added this line for clarity

Data from the entire campaign was used to perform the correlation analysis

7. *Lines 332-333: The authors suggest that the concentrations of BC, NOx, SO2, and CO could be influenced by meteorological conditions, which might explain the moderate correlation observed between AlkOA and these pollutants. However, given that AlkOA is measured as part of AMS sub-micron samples, it is reasonable to assume that AlkOA would also be influenced*

*by meteorological factors. Additionally, since BC is also in the particle phase, the influence of meteorological conditions on both AlkOA and BC should be similar.*

Response: Thank you for your feedback. These lines been added to the manuscript for more clarity.

Similarly, AlkOA could potentially be affected by meteorological conditions. Since AlkOA is measured as part of AMS sub-micron particles, it is expected to behave similarly to eBC in the particle phase. Therefore, meteorological conditions likely influence both AlkOA and eBC in a similar manner.

8. *Figures 5 and 6: The authors did not address the previous question about the significant decrease in AlkOA from hour 23 to hour 0, which could be associated with the large standard deviation when averaging the data to study the diurnal variation. I recommend that the authors consider the standard deviation when discussing the diurnal variation if it is large.*

Response: Thank you for the suggestion. Based on the reviewer's comment, Figures 5 and 6, which display the mean diurnal pattern along with a measure of variability, have been updated. These lines have been added to the manuscript for greater clarity.

The increase in AlkOA concentration from 22:00 to 23:00, or the subsequent decrease from 23:00 to 00:00, falls within the variability range of the 00:00 to 01:00 period. Therefore, a statistically significant decrease in AlkOA concentration from 23:00 to 00:00 is hardly measurable. Meteorological factors may contribute to the variability in the diurnal cycle observed during this period. Additionally, unidentified local source such as airport ground service equipment could potentially explain the variability observed from 22:00 to 00:00.

[Figure]

**Figure 5 shows the mean diurnal patterns of LO-OOA, AlkOA, and MO-OOA during the measurement campaign, with a 95% confidence interval**.

**Figure 6 shows the normalized level of AlkOA and trace gases during the measurement campaign, with a 95% confidence interval**

---

## Author Response (AR3)

We appreciate your valuable time in reviewing this manuscript. The time has been corrected as shown in Figure 5.

**Editor**

1.  *Thanks for addressing the reviewer's comments and clarifying the questions. Looking at Figure 5, I disagree that AlkOA concentration was increasing between 9 am-8 pm. The period with an obvious increase is 9 am-6 pm. However, there is a clear drop in the concentration at 6 pm; therefore, I'd like you to revise the description of this Figure in Lines 332-335 to accurately represent the data.*

    Response: Thank you for pointing this out. The description of this Figure has been adjusted to the correct time.